# Integrating Mixed Livestock Systems to Optimize Forage Utilization and Modify Woody Species Composition in Semi-Arid Communal Rangelands

Mhlangabezi Slayi [1,*] and Ishmael Festus Jaja [2]

1   Centre for Global Change (CGC), Faculty of Science and Agriculture, University of Fort Hare,
    Alice 5700, South Africa
2   Department of Livestock and Pasture Science, Faculty of Science and Agriculture, University of Fort Hare,
    Alice 5700, South Africa; ijaja@ufh.ac.za
*   Correspondence: mslayi@ufh.ac.za

**Abstract:** Communally owned rangelands serve as critical grazing areas for mixed livestock species such as cattle and goats, particularly in the arid and semi-arid regions of Southern Africa. This study aimed to evaluate the nutritional composition and woody species composition of communal rangelands where cattle and goat flocks graze together and to investigate the influence of grazing intensity on vegetation dynamics. Vegetation surveys were conducted across varying grazing intensities to assess species richness, biomass, and dietary preferences, while soil properties were analyzed to determine their interaction with vegetation attributes. Stepwise regression and path analyses were used to explore the relationships between soil characteristics, vegetation structure, and livestock dietary choices. The results revealed that high grazing pressure significantly reduced grass biomass ($p = 0.003$) and woody species density ($p = 0.007$) while increasing shrub cover ($p = 0.018$). Nutritional analysis indicated that goats preferred woody shrubs, which contributed 42.1% of their diet compared to 27.8% for cattle ($p = 0.008$). Regression analysis further showed that soil organic carbon ($p = 0.002$) and tree height ($p = 0.041$) were strong predictors of shrub cover. Seasonal variation significantly affected forage availability and nutritional content, with higher crude protein levels recorded during the wet season ($p = 0.007$). These findings suggest that grazing management strategies should be tailored to the distinct forage needs of cattle and goats to maintain the productivity and ecological stability of communal rangelands. A holistic approach that considers livestock dietary preferences, vegetation composition, and soil health is essential for sustainable rangeland management in mixed-species grazing systems.

**Keywords:** forage selection; co-grazing; goats; ruminant nutrition; communal farming; bush encroachment

## 1. Introduction

Communally managed rangelands are vital for the livelihoods of rural communities in Sub-Saharan Africa [1,2], providing essential resources such as grazing land, fuelwood, and various ecosystem services [3,4]. These rangelands typically feature the co-grazing of multiple livestock species, especially cattle and goats, each exhibiting distinct grazing behaviors and dietary preferences [5,6]. Cattle primarily feed on grasses, whereas goats are more adaptable browsers, consuming a wide array of herbaceous and woody plants [7,8]. This coexistence influences not only forage availability but also the overall structure and species composition of rangeland vegetation [9,10].

Understanding the nutritional quality and composition of woody species in rangelands is essential for sustainable management, as variations in forage quality and plant diversity greatly affect animal performance and ecological integrity [11,12]. The nutritional quality of forage is a critical determinant of livestock productivity, impacting growth rates,

reproductive success, and resilience to environmental stressors [13,14]. In rangeland systems with limited feed resources, it is crucial to maintain a balance between herbaceous and woody vegetation to meet the dietary needs of mixed livestock herds [15,16]. However, the communal nature of these rangelands presents unique challenges in reconciling livestock demands with vegetation sustainability [17,18]. Overgrazing and selective browsing by goats can shift plant species composition, often leading to the dominance of less palatable or invasive woody species [19–21]. Such transformations not only decrease the availability of high-quality forage for cattle but can also contribute to land degradation and biodiversity loss [22,23].

Despite the expanding body of research on rangeland dynamics, few studies have thoroughly examined the combined effects of mixed grazing on the nutritional quality and woody species composition of communally owned rangelands [24,25]. This study aims to address this gap by assessing the nutritional profiles of key forage species and analyzing the spatial and temporal distribution patterns of woody species in communal rangelands where cattle and goats graze together. By identifying species preferences, dietary overlaps, and potential resource competition, the findings can guide targeted management interventions to enhance livestock productivity while preserving rangeland health. Ultimately, understanding the complex interactions between livestock species and vegetation in these shared systems is essential for developing sustainable grazing strategies that promote both livestock productivity and ecological resilience.

## 2. Materials and Methods

### 2.1. Ethical Considerations

This study was conducted in accordance with ethical guidelines for animal research and received approval (JAJ051SMPO01) from the institutional Animal Research Ethics Committee. All animal handling and observation procedures were designed to minimize stress and disturbance to the animals.

### 2.2. Study Area Description

This study was conducted in the communal rangelands of Middledrift, a semi-arid region in the Eastern Cape Province of South Africa (Figure 1). This area experiences a bimodal rainfall pattern, receiving approximately 412 mm of rain per year, with the majority falling during autumn. July typically sees the lowest rainfall, at 9 mm, while March records the highest, at 66 mm. The monthly distribution of average daily maximum temperatures indicates that midday temperatures in Middledrift range from 19.6 °C in June to 27.3 °C in February. The region is coldest in July, with average nighttime temperatures dropping to 6 °C. The topography is characterized by gentle slopes interspersed with small hills, and the soils are primarily sandy loams with varying levels of fertility [26]. The vegetation consists of a mix of herbaceous species and woody plants typical of savanna ecosystems, including *Acacia*, *Combretum*, and *Terminalia* species [27,28]. Land use in the area is dominated by livestock production, with cattle and goats being the primary species grazed by local communities. The communal rangeland in Middledrift spans approximately 15,000 hectares, divided into multiple grazing zones managed by local community cooperatives. These zones are not uniformly managed, leading to variations in forage availability and grazing pressure. The typical livestock composition comprises roughly 60% cattle and 40% goats, reflecting the dual-purpose approach of the local farming systems. Cattle are primarily valued for meat and draft purposes, while goats contribute to meat, milk, and skin production. The shared grazing areas often experience seasonal variability in stocking density, influenced by climatic conditions and communal management practices.

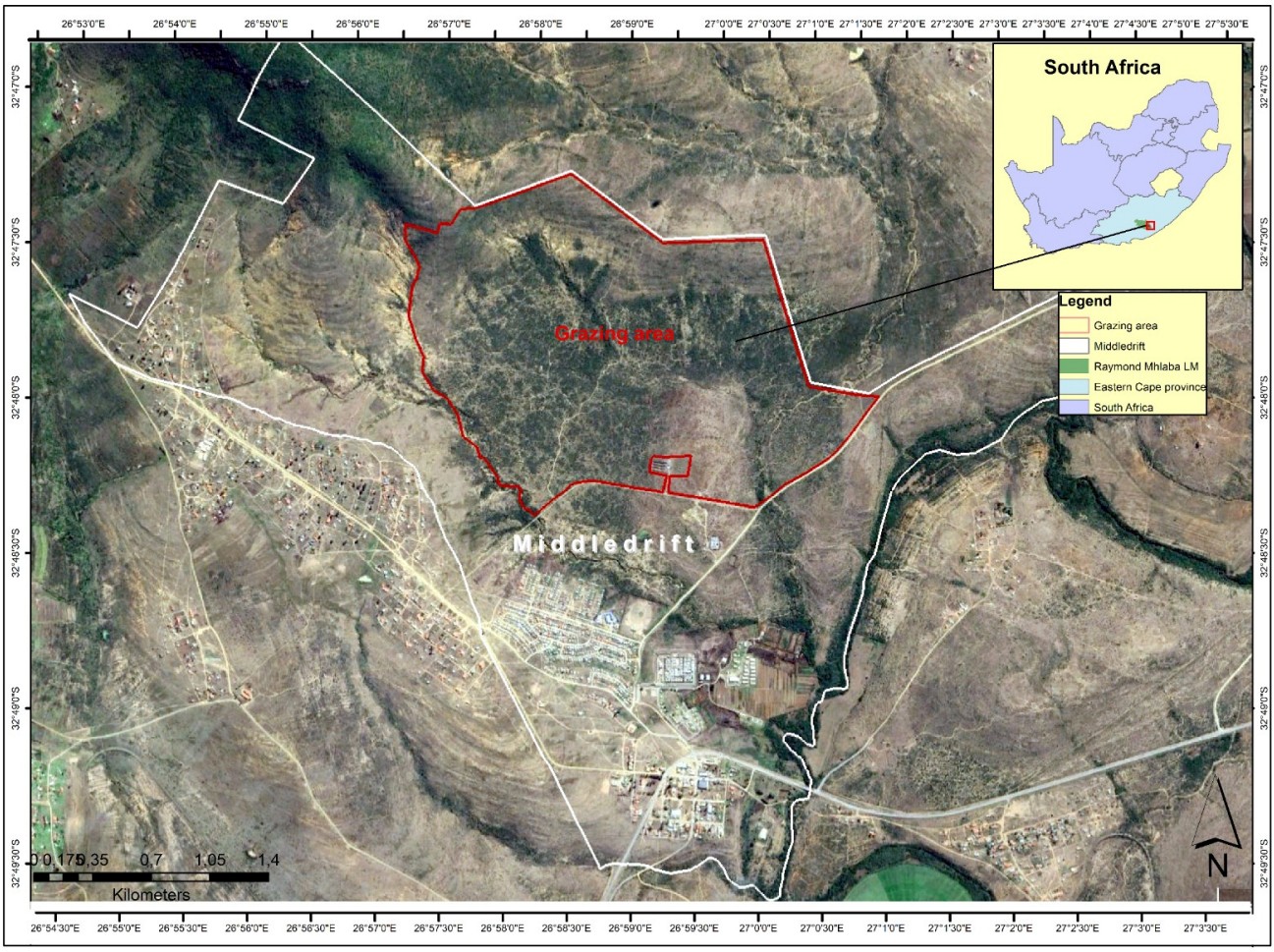

**Figure 1.** Map showing the location of the study site.

*2.3. Experimental Design and Sampling Protocol*

　　To investigate the nutritional quality and woody species composition of communal rangelands, a stratified sampling approach was used to capture variability across grazing zones with differing levels of use. The grazing zones in this study have been managed in a similar manner for over 50 years, allowing for consistent patterns of grazing impact across areas. The study was conducted in three distinct grazing areas—heavy use, moderate use, and light use—each selected based on proximity to watering points, which are key determinants of grazing intensity. These zones are spatially dispersed across the rangelands but share similar vegetation communities and soil types, ensuring the comparability of findings across environmental conditions. Heavy-use areas, located near watering points, experience the highest grazing pressure due to the frequent congregation of livestock, often resulting in reduced vegetation cover, altered species composition, and potentially lower forage quality. In contrast, moderate-use areas, positioned midway between watering points and the periphery, face moderate grazing pressure, providing a balanced representation of grazing impacts. Light-use areas, found at the outermost edges of the rangelands, experience minimal grazing pressure, as livestock rarely reach these zones, leading to denser and more diverse vegetation structures with potentially higher-quality forage. This study controls for non-grazing-related environmental differences by selecting zones within similar vegetation and soil types, thus isolating grazing intensity as the primary variable. Additionally, sampling in each zone was conducted during both wet and dry seasons to account for seasonal variations in forage availability and species composition. This approach provides a comprehensive view of how both grazing intensity and seasonal changes influence vegetation structure and forage quality across communal rangelands.

### 2.3.1. Site Selection

A total of nine sampling sites (three per grazing zone) were randomly selected within each zone based on accessibility and vegetation characteristics. Each sampling site measured 50 m × 50 m, ensuring adequate coverage of plant diversity.

### 2.3.2. Vegetation Assessment

Herbaceous Layer: To assess the herbaceous layer, a modified step-point method was used along 3 transects (50 m each) per site. At 2 m intervals along each transect, the nearest herbaceous plant species was identified and its canopy cover recorded. In addition to canopy cover, the presence of bare ground was also recorded at each point, as it serves as an important indicator of grazing intensity and soil exposure. The dry matter biomass was estimated using a quadrat (0.5 m × 0.5 m) placed randomly at 5 points per transect, and clipped samples were oven-dried at 65 °C for 48 h to determine biomass. By measuring both vegetation cover and bare ground, this approach provides a comprehensive view of herbaceous layer conditions and the impact of grazing on ground cover.

### 2.3.3. Woody Species Composition

Woody species composition was assessed using the point-centered quarter method along the same transects. At 10 m intervals, the distance to the nearest woody plant (≥0.5 m height) in each quarter was recorded, along with species identification and height measurement. Basal area, crown cover, and woody species density were calculated using standard formulas.

### *2.4. Forage Collection for Nutritional Analysis*

Representative samples of commonly grazed forage species, including both herbaceous and woody plants, were collected from each study site. Herbaceous species were clipped at ground level, while leaf and small stem samples were selectively harvested from woody species [29,30]. The collected samples were then pooled for each species within a site to ensure uniformity, air-dried to maintain sample integrity, and subsequently ground to pass through a 1 mm sieve for precise analysis [31,32]. The processed samples were analyzed for key nutritional parameters. Crude protein (CP) content was determined using the Kjeldahl method, a standard approach for measuring total nitrogen in feed. Fiber content, including neutral detergent fiber (NDF) and acid detergent fiber (ADF), was assessed using the Van Soest method, which provides insight into the digestibility and energy value of forages. Metabolizable energy (ME), crucial for evaluating the potential energy available to livestock, was calculated using the in vitro gas production technique. Additionally, mineral content was quantified using Atomic Absorption Spectrophotometry, which accurately measured concentrations of essential minerals such as calcium (Ca), phosphorus (P), and magnesium (Mg). This comprehensive nutritional profiling enabled a detailed understanding of the quality and suitability of various forage species for livestock feeding across different grazing zones.

### *2.5. Animal Grazing Behavior and Diet Selection*

To assess the grazing behavior of and diet selection by cattle and goats, a combination of direct observation and fecal analysis was employed. Five mature animals of each species were randomly selected from mixed herds for focal sampling during both morning and evening grazing periods. Each animal was observed for a 2 h period on a rotational basis, and feeding behaviors, such as grazing versus browsing, along with plant species selection, were recorded every 5 min. This focal sampling technique allowed calculation of the proportion of time spent on different plant functional groups (grasses, forbs, shrubs, and trees), as well as the number of bites per plant species, to determine overall diet composition. Complementing the observations, fecal samples were collected from the same animals at the end of each observation period and analyzed using microhistological techniques, a method widely documented in previous studies on diet composition analysis in grazing

animals [33–35]. This technique allows for the identification and quantification of plant fragments in fecal matter, providing an accurate estimate of the proportion of different forage species consumed by cattle and goats. Microhistological analysis is preferred in this context due to its effectiveness in reflecting diet diversity and plant selection in herbivores over time, particularly in complex rangeland systems where direct observation of intake is challenging.

*2.6. Data Analysis*

Vegetation data were collected to analyze species composition and diversity, with woody and herbaceous species evaluated using relative frequency and the Importance Value Index (IVI). Species diversity was further assessed through the Shannon–Weiner and Simpson's diversity indices to capture variations in biodiversity across different grazing zones and seasons. Soil type was also included in the analysis to account for its potential influence on vegetation characteristics and grazing impacts, as soil properties can affect plant growth and nutrient availability. The soil types across treatments were sandy loam and silt loam, selected to ensure that they represented the typical soil variability of the study area while allowing for consistent comparison across grazing zones. Herbaceous biomass and woody plant density were examined through a two-way Analysis of Variance (ANOVA) to determine the impact of grazing zone and season, followed by Tukey's HSD post hoc tests for detailed pairwise comparisons. For nutritional analysis, mean values of crude protein (CP), neutral detergent fiber (NDF), acid detergent fiber (ADF), and metabolizable energy (ME) for each forage species were compared using one-way ANOVA, while non-normally distributed data were analyzed using the Kruskal–Wallis test. Pearson's correlation analysis was applied to identify potential relationships between forage quality and livestock dietary preferences, and grazing behavior and diet selection were assessed by analyzing the percentage of time cattle and goats spent grazing, browsing, and feeding on different species, with Chi-square tests used to detect species differences. Pianka's Index was used to calculate diet overlap, measuring competition for forage resources. To understand the broader interactions between grazing patterns and vegetation attributes, Canonical Correspondence Analysis (CCA) was conducted, capturing relationships between vegetation variables, soil types, and environmental gradients across grazing zones. Additionally, path analysis was beneficial in exploring causal relationships between variables such as grazing pressure, vegetation diversity, and livestock diet composition, clarifying direct and indirect effects within ecosystem dynamics. Integrating these methods provided a comprehensive understanding of how grazing patterns, soil types, and vegetation composition interact to affect forage quality across different grazing zones and seasons.

## 3. Results

### 3.1. Herbaceous and Woody Species Composition and Abundance Across Grazing Zones

Table 1 illustrates a clear variation in species composition and abundance across the heavy, moderate, and light grazing zones, reflecting the differential impacts of grazing intensity on plant communities. The heavy-use zone is predominantly characterized by herbaceous grasses, such as *Cenchrus ciliaris* and *Eragrostis curvula*, which exhibit high relative frequency (45.3% and 20.2%, respectively) and density (3500 and 1700 plants/ha). In contrast, woody species like *Acacia karroo* and *Dichrostachys cinerea* are less abundant, with lower relative frequency values (12.1% and 7.5%, respectively), reflecting a grazing regime that limits the establishment and growth of woody species. Although these shrubs have notable basal areas (23.1 cm$^2$/plant and 15.6 cm$^2$/plant) and moderate heights (160 to 185 cm), their reduced density and frequency suggest that intense herbivory and competition from grasses might be inhibiting their proliferation. In the moderate-use zone, the plant community comprises a more balanced mixture of grasses and woody species. Dominant grasses such as *Themeda triandra* (39.5%) and *Panicum maximum* (25.7%) are accompanied by key shrubs like *Combretum hereroense* and *Grewia occidentalis*, which have a relatively high Importance Value Index (IVI) and contribute significantly to structural diversity.

**Table 1.** Herbaceous and woody species composition and abundance across grazing zones.

| Grazing Zone | Species Name | Functional Group | Relative Frequency (%) | Importance Value Index (IVI) | Density (Plants/ha) | Basal Area (cm$^2$/Plant) | Height (cm) |
|---|---|---|---|---|---|---|---|
| Heavy Use | *Cenchrus ciliaris* | Grass | 45.3 | 0.67 | 3500 | - | - |
| Heavy Use | *Acacia karroo* | Shrub | 12.1 | 0.85 | 450 | 23.1 | 185 |
| Heavy Use | *Eragrostis curvula* | Grass | 20.2 | 0.49 | 1700 | - | - |
| Heavy Use | *Dichrostachys cinerea* | Shrub | 7.5 | 0.60 | 320 | 15.6 | 160 |
| Moderate Use | *Themeda triandra* | Grass | 39.5 | 0.72 | 2800 | - | - |
| Moderate Use | *Panicum maximum* | Grass | 25.7 | 0.55 | 2200 | - | - |
| Moderate Use | *Combretum hereroense* | Shrub | 13.2 | 0.77 | 540 | 27.8 | 240 |
| Moderate Use | *Grewia occidentalis* | Shrub | 10.8 | 0.70 | 480 | 21.2 | 200 |
| Light Use | *Terminalia sericea* | Tree | 10.4 | 1.01 | 180 | 47.2 | 340 |
| Light Use | *Brachystegia spiciformis* | Tree | 8.9 | 0.89 | 160 | 50.1 | 380 |
| Light Use | *Setaria sphacelata* | Grass | 35.8 | 0.65 | 2300 | - | - |
| Light Use | *Peltophorum africanum* | Tree | 12.7 | 0.95 | 200 | 42.5 | 310 |

### 3.2. Nutritional Composition of Major Forage Species

The nutritional composition of the major forage species, as presented in Table 2, reveals distinct variations among grasses, shrubs, and tree species, highlighting their differential nutritional quality and potential contributions to the diets of grazing livestock. Among the grasses, *Themeda triandra* and *Panicum maximum* stand out with higher crude protein (CP) levels of 9.2% and 9.7%, respectively, compared to other grasses like *Cenchrus ciliaris* (7.5%) and *Eragrostis curvula* (8.0%). This higher protein content, along with relatively lower neutral detergent fiber (NDF) and acid detergent fiber (ADF) values, suggests that these grasses provide better-quality forage, with higher digestibility and energy content (9.1–9.8 MJ/kg of metabolizable energy). The mineral composition of these grasses is comparatively low, with calcium and phosphorus values ranging from 0.12% to 0.19% and 0.05% to 0.09%, respectively, indicating that they may require supplementation, especially in nutrient-deficient grazing systems. In contrast, the shrubs and tree species demonstrate superior nutritional quality in several aspects. *Acacia karroo*, *Dichrostachys cinerea*, and *Combretum hereroense* show higher CP content (11.8–13.5%) and lower fiber fractions (NDF: 46.3–49.6%; ADF: 26.8–29.4%) compared to the grasses. This nutritional profile indicates higher palatability and energy availability, making them important sources of protein and energy, particularly during the dry season when grass quality typically declines. The metabolizable energy content of these shrubs is also relatively high, ranging from 10.4 to 11.3 MJ/kg, which can sustain livestock during periods of forage scarcity.

Moreover, these shrubs have markedly higher calcium and phosphorus levels (0.10–1.25%), contributing essential minerals that support bone development and metabolic processes in grazing animals. *Combretum hereroense*, with the highest calcium (1.25%) and phosphorus (0.14%) levels among all the species, emerges as a particularly valuable component of the forage base, capable of meeting some of the mineral requirements of livestock. Tree species such as *Terminalia sericea*, *Peltophorum africanum*, and *Brachystegia spiciformis* present intermediate nutritional values compared to shrubs and grasses. They exhibit moderate CP content (9.4–11.2%), NDF (51.5–55.2%), and ADF (30.0–31.2%) values, with metabolizable energy content ranging from 9.0 to 9.9 MJ/kg. Their relatively high calcium

levels (0.85–0.95%) and phosphorus concentrations (0.09–0.11%) suggest that these trees can play a supplementary role in balancing the nutritional deficiencies of herbaceous species, especially in areas where overgrazing has depleted the grass layer.

**Table 2.** Nutritional composition of major forage species (mean ± SE).

| Species Name | Type | Crude Protein (%) | NDF (%) | ADF (%) | Metabolizable Energy (MJ/kg) | Calcium (%) | Phosphorus (%) |
|---|---|---|---|---|---|---|---|
| *Cenchrus ciliaris* | Grass | 7.5 ± 0.3 | 65.4 ± 1.2 | 35.7 ± 0.9 | 8.2 ± 0.5 | 0.15 ± 0.02 | 0.06 ± 0.01 |
| *Themeda triandra* | Grass | 9.2 ± 0.4 | 58.9 ± 0.8 | 32.1 ± 0.7 | 9.1 ± 0.4 | 0.19 ± 0.03 | 0.07 ± 0.02 |
| *Acacia karroo* | Shrub | 12.7 ± 0.5 | 48.5 ± 1.0 | 29.4 ± 0.8 | 10.6 ± 0.3 | 1.22 ± 0.05 | 0.12 ± 0.02 |
| *Terminalia sericea* | Tree | 10.1 ± 0.3 | 52.3 ± 1.1 | 30.9 ± 0.6 | 9.4 ± 0.3 | 0.85 ± 0.04 | 0.11 ± 0.01 |
| *Eragrostis curvula* | Grass | 8.0 ± 0.2 | 62.7 ± 1.3 | 34.6 ± 0.9 | 8.6 ± 0.5 | 0.12 ± 0.02 | 0.05 ± 0.01 |
| *Panicum maximum* | Grass | 9.7 ± 0.4 | 59.4 ± 1.2 | 31.5 ± 0.7 | 9.8 ± 0.4 | 0.17 ± 0.03 | 0.09 ± 0.02 |
| *Dichrostachys cinerea* | Shrub | 11.8 ± 0.3 | 49.6 ± 1.0 | 28.9 ± 0.8 | 10.4 ± 0.4 | 1.15 ± 0.06 | 0.10 ± 0.02 |
| *Grewia occidentalis* | Shrub | 12.3 ± 0.4 | 47.8 ± 1.1 | 27.6 ± 0.7 | 11.0 ± 0.3 | 1.09 ± 0.05 | 0.13 ± 0.03 |
| *Setaria sphacelata* | Grass | 8.5 ± 0.2 | 60.1 ± 0.9 | 33.3 ± 0.6 | 8.9 ± 0.4 | 0.14 ± 0.03 | 0.06 ± 0.01 |
| *Peltophorum africanum* | Tree | 11.2 ± 0.3 | 51.5 ± 1.2 | 30.0 ± 0.7 | 9.9 ± 0.5 | 0.95 ± 0.04 | 0.10 ± 0.02 |
| *Combretum hereroense* | Shrub | 13.5 ± 0.4 | 46.3 ± 1.2 | 26.8 ± 0.7 | 11.3 ± 0.5 | 1.25 ± 0.05 | 0.14 ± 0.03 |
| *Brachystegia spiciformis* | Tree | 9.4 ± 0.3 | 55.2 ± 0.8 | 31.2 ± 0.5 | 9.0 ± 0.4 | 0.89 ± 0.03 | 0.09 ± 0.01 |

Neutral detergent fiber (NDF); acid detergent fiber (ADF).

### 3.3. Seasonal Variation in Biomass and Cover of Herbaceous Vegetation

Figure 2 illustrates the impact of grazing intensity and seasonal changes on herbaceous biomass, canopy cover, species richness, and the Shannon–Weiner diversity index across heavy-use, moderate-use, and light-use grazing zones. Herbaceous biomass is notably higher in light-use zones across all seasons, peaking at 1650 kg/ha in the wet season, whereas heavy-use zones consistently exhibit lower biomass, especially during the dry season, where it falls to 380 kg/ha. This trend suggests that reduced grazing pressure promotes vegetation growth, which is most pronounced in favorable conditions in the wet season. Canopy cover follows a similar pattern, with light-use areas maintaining the densest canopy in all seasons (up to 72.8% in the wet season), while heavy-use zones show a substantial reduction, particularly in the dry season (32.1%), likely due to vegetation depletion from intensive grazing. Species richness and diversity also vary with grazing intensity, as light-use zones support a greater variety of species, achieving the highest richness (25 species) and diversity index (2.12) in the wet season. Conversely, heavy-use areas show lower species richness and diversity, dropping to 10 species and an index of 1.12 in the dry season. This pattern indicates that intense grazing limits species diversity, potentially by favoring grazing-resistant plants that dominate the community. Across all metrics, the wet season and lower grazing intensity support greater biomass, cover, species richness, and diversity, while heavy grazing and the dry season are associated with reduced vegetation growth and diversity, likely due to combined pressures of resource scarcity and grazing stress.

### 3.4. Woody Species Characteristics and Utilization by Cattle and Goats

Table 3 outlines the characteristics and utilization patterns of woody species across various grazing zones, highlighting differences in plant morphology, browsing intensity, and the preferences of primary browsers. In light-use zones, woody species typically exhibit superior structural attributes, including greater plant heights, basal areas, and crown cover, compared to those found in moderate- and heavy-use zones. For example, *Vachellia tortilis* in the light-use zone has the tallest average height (400 cm), the highest basal area (68.9 cm$^2$/plant), and extensive crown cover (78.3%), along with the lowest

browsing pressure (15%) and a utilization index of 0.20, indicating minimal impact from herbivory. Other species in this zone, such as *Terminalia sericea* and *Brachystegia spiciformis*, display similar trends with lower browsing percentages (30% and 20%, respectively), primarily utilized by cattle. This suggests that the reduced grazing pressure in light-use zones facilitates healthier growth and less intense utilization of woody species. Conversely, heavy-use zones are marked by shorter plant heights and diminished basal areas due to the adverse effects of intensive browsing pressure, mainly from goats. Species like *Dichrostachys cinerea* and *Senegalia mellifera* experience the highest browsing percentages (85% and 80%, respectively) and exhibit high utilization indices (0.92 and 0.87), indicating significant exploitation. The reduced crown cover of these species (25.6% and 28.4%, respectively) further illustrates the impact of frequent browsing, likely stunting growth and compromising overall plant health.

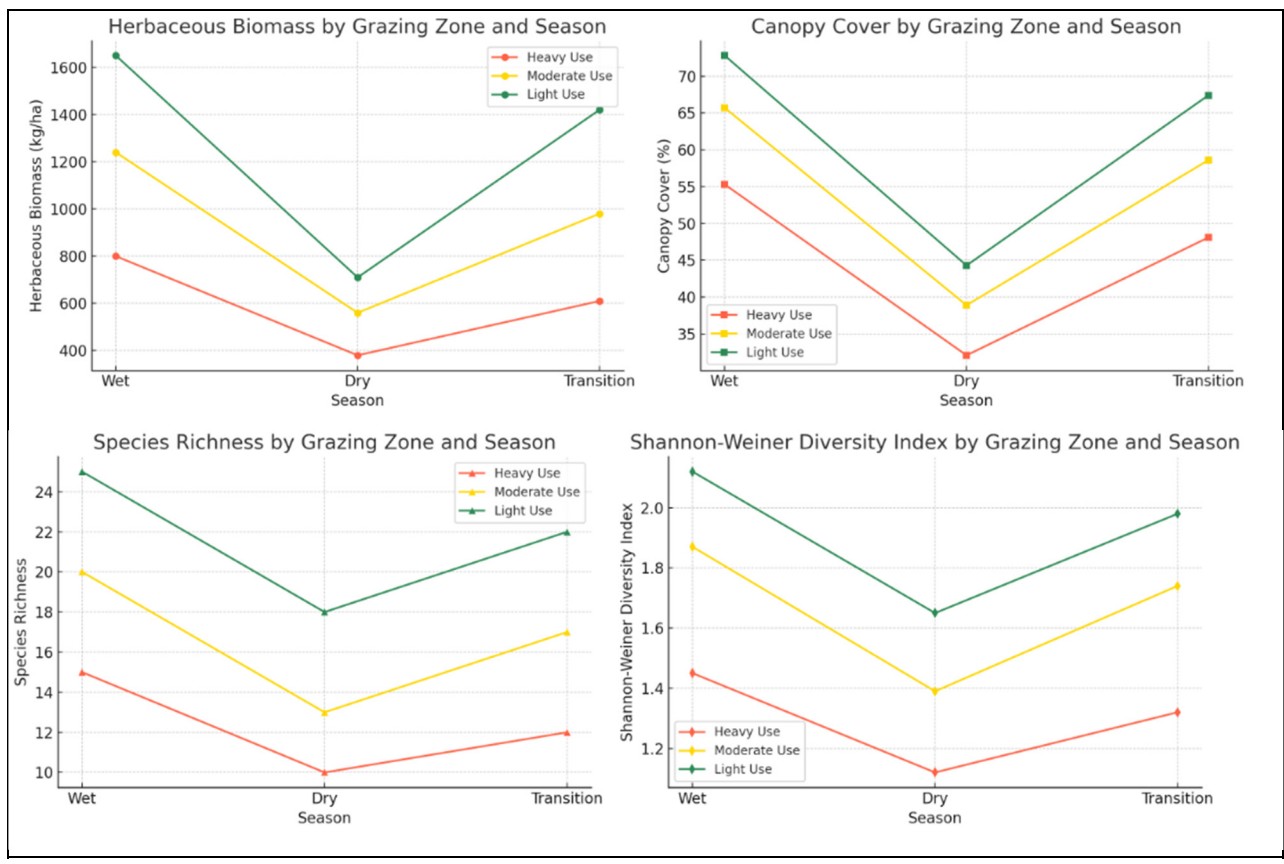

**Figure 2.** Comparison of herbaceous biomass, canopy cover, species richness, and Shannon–Weiner diversity index across grazing zones and seasons.

Similarly, *Acacia karroo* in the heavy-use zone experience a browsing percentage of 75% and a utilization index of 0.85, highlighting goats' preference for this species. Moderate-use zones display intermediate characteristics, with species like *Combretum hereroense* and *Ziziphus mucronata* showing moderate plant heights (250 cm and 260 cm) and basal areas (42.7 cm$^2$ and 41.9 cm$^2$, respectively). Browsing intensity is relatively balanced, ranging between 45% and 60%, with utilization indices between 0.68 and 0.72. Both goats and cattle browse these species, indicating a shared preference likely linked to the balanced grazing pressure in these zones. Species such as *Grewia flava* and *Rhus lancea* also demonstrate similar utilization patterns, being moderately browsed by cattle (50%). This suggests that moderate-use zones can support a diverse array of woody species without intense overexploitation.

**Table 3.** Woody species characteristics and utilization by cattle and goats.

| Species Name | Grazing Zone | Plant Height (cm) | Basal Area (cm²/Plant) | Crown Cover (%) | Percentage Browsed (%) | Primary Browser (Cattle/Goats) | Utilization Index |
|---|---|---|---|---|---|---|---|
| *Acacia karroo* | Heavy Use | 180 | 35.2 | 45.1 | 75 | Goats | 0.85 |
| *Combretum hereroense* | Moderate Use | 250 | 42.7 | 38.2 | 60 | Goats | 0.72 |
| *Terminalia sericea* | Light Use | 320 | 53.8 | 65.4 | 30 | Cattle | 0.35 |
| *Dichrostachys cinerea* | Heavy Use | 120 | 28.4 | 25.6 | 85 | Goats | 0.92 |
| *Grewia flava* | Moderate Use | 240 | 40.3 | 50.7 | 50 | Cattle | 0.64 |
| *Brachystegia spiciformis* | Light Use | 380 | 57.5 | 70.2 | 20 | Cattle | 0.28 |
| *Peltophorum africanum* | Light Use | 310 | 49.8 | 62.1 | 35 | Cattle | 0.40 |
| *Albizia harveyi* | Moderate Use | 270 | 45.6 | 55.3 | 45 | Goats | 0.68 |
| *Senegalia mellifera* | Heavy Use | 140 | 33.7 | 28.4 | 80 | Goats | 0.87 |
| *Ziziphus mucronata* | Moderate Use | 260 | 41.9 | 47.5 | 55 | Cattle | 0.70 |
| *Sclerocarya birrea* | Light Use | 360 | 61.3 | 75.8 | 25 | Cattle | 0.32 |
| *Euclea divinorum* | Heavy Use | 150 | 30.5 | 35.2 | 65 | Goats | 0.78 |
| *Rhus lancea* | Moderate Use | 280 | 44.2 | 53.6 | 50 | Cattle | 0.67 |
| *Vachellia tortilis* | Light Use | 400 | 68.9 | 78.3 | 15 | Cattle | 0.20 |

*3.5. Livestock Diet Overlap and Resource Competition Index*

Figure 3 illustrates the relationships between dietary overlap, species shared, and competition index (CI) across different grazing zones and seasons. Pianka's Index of Dietary Overlap shows higher values in heavily used grazing zones across all seasons, indicating a significant overlap in diet among herbivores, particularly in the dry season (0.75) and wet–dry transitional season (0.70). This dietary overlap suggests heightened resource sharing, especially when forage is limited. Similarly, the percentage of species shared tends to be highest in heavy-use zones, peaking at 55% during the dry season, reflecting a reduced diversity of forage resources in heavily grazed areas. As grazing intensity decreases from heavy to light use, Pianka's Index, species shared, and competition index all generally decrease. This trend is most pronounced in the riparian zone, where values are lowest across all metrics, indicating reduced dietary overlap and competition likely due to greater forage availability and variety. The competition index follows a similar seasonal and grazing intensity pattern, reaching its maximum in heavy-use zones (0.82 during the dry season) and its minimum in the riparian zone (0.46 during the transitional season). These patterns suggest that grazing pressure intensifies resource competition, especially under limited forage conditions in heavy-use areas, and highlight the role of riparian and transitional zones in potentially reducing dietary overlap and competition among herbivores due to more diverse and abundant forage availability.

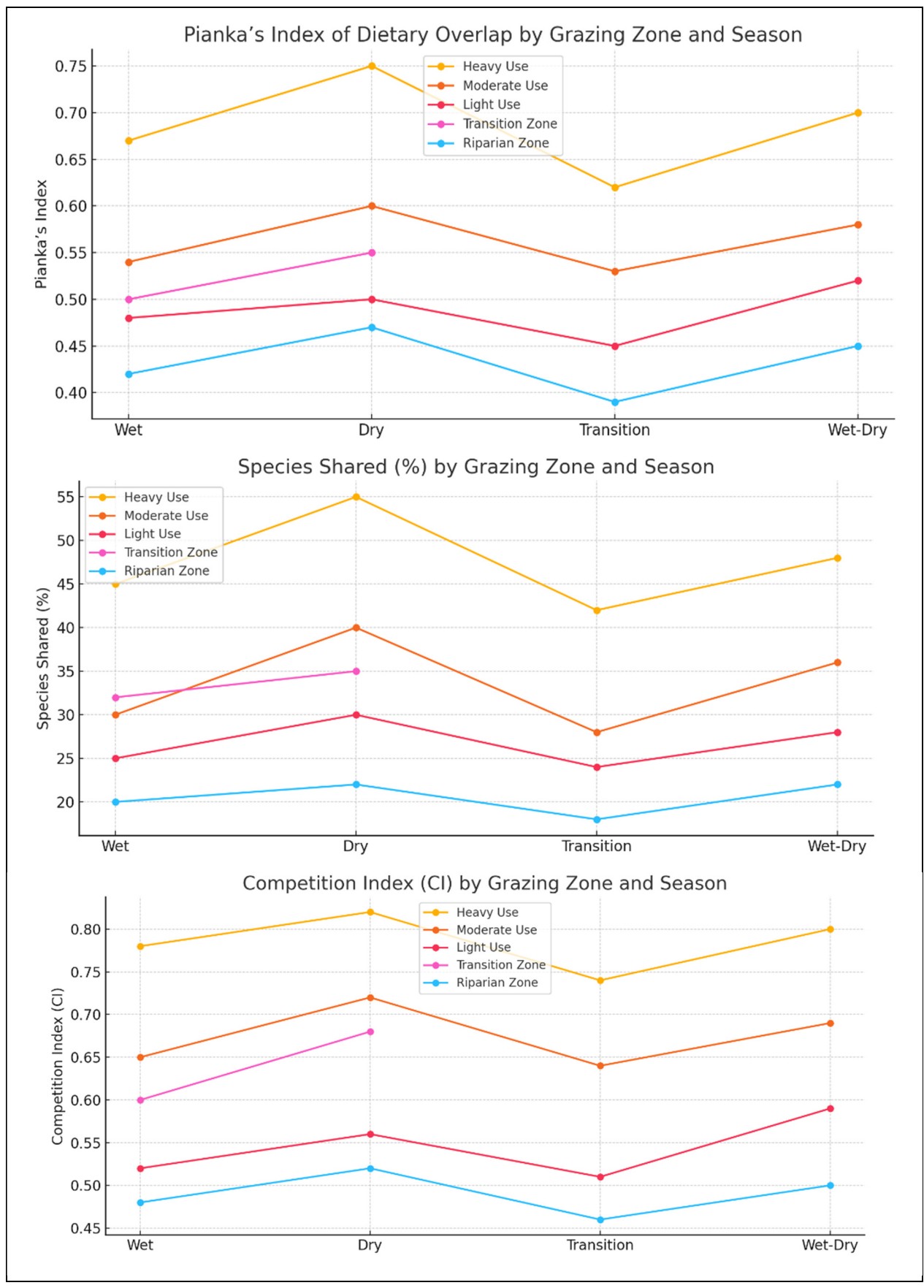

**Figure 3.** Dietary overlap, species shared, and competition index across grazing zones and seasons.

*3.6. Seasonal Changes in Woody Species Recruitment and Mortality Rates*

Table 4 illustrates the seasonal fluctuations in the recruitment and mortality rates of woody species across various grazing zones, highlighting distinct patterns of net woody regeneration influenced by grazing pressure. In heavy-use zones, species such as *Acacia karroo*, *Dichrostachys cinerea*, and *Senegalia mellifera* exhibit negative net woody regeneration rates, as mortality exceeds recruitment. For example, *Acacia karroo* has a recruitment rate of 50 seedlings per hectare but a mortality rate of 75 plants per hectare, resulting in a net loss of 25 plants per hectare. This trend was examined across both wet and dry seasons to capture any seasonal fluctuations in regeneration rates. While recruitment rates for some species showed slight increases during the wet season, mortality rates remained consistently high across both seasons, reinforcing the role of heavy grazing pressure in limiting woody regeneration regardless of seasonal changes. Similar patterns are observed in other species within this zone, indicating that heavy grazing pressure significantly hinders the regeneration of woody vegetation, which may lead to long-term declines in vegetation. In contrast, moderate-use zones display positive net regeneration rates for woody species, reflecting a balance between recruitment and mortality. Species such as *Terminalia sericea* and *Peltophorum africanum* show net increases of 50 and 45 plants per hectare, respectively, suggesting that moderate grazing pressure fosters the establishment and persistence of woody plants. These zones demonstrate relatively healthy regeneration dynamics, which could contribute to a stable woody vegetation structure over time. Light-use zones feature markedly higher recruitment rates and lower mortality rates, resulting in significant net woody regeneration. For example, *Combretum hereroense* and *Sclerocarya birrea* display net increases of 75 and 87 plants per hectare, respectively, indicating strong regeneration and minimal grazing impact. This combination of high recruitment and low mortality fosters a favorable environment for the proliferation of woody species, promoting a denser and more diverse woody vegetation structure. Riparian zones exhibit the highest net woody regeneration rates among all zones, with species such as *Vachellia tortilis* and *Faidherbia albida* achieving net increases of 95 and 98 plants per hectare, respectively. These riparian areas are located adjacent to perennial water sources, typically at the boundaries of the grazing zones. Although the heavy-use zones are positioned near artificial watering points to attract livestock, they are generally distinct from riparian zones, which maintain a natural water flow and vegetation buffer. This separation likely contributes to the enhanced regeneration rates observed in riparian zones, as they experience reduced grazing intensity and benefit from consistent water availability, supporting a favorable environment for woody species recruitment. The favorable moisture conditions and low grazing pressure in these areas likely enhance seedling establishment and survival, making riparian zones vital for maintaining woody species diversity and providing ecological refuges in heavily grazed landscapes.

**Table 4.** Seasonal changes in woody species recruitment and mortality rates.

| Grazing Zone | Species Name | Recruitment Rate (Seedlings/ha) | Mortality Rate (Plants/ha) | Net Woody Regeneration (Plants/ha) |
|---|---|---|---|---|
| Heavy Use | *Acacia karroo* | 50 | 75 | −25 |
| Heavy Use | *Dichrostachys cinerea* | 30 | 60 | −30 |
| Heavy Use | *Senegalia mellifera* | 45 | 70 | −25 |
| Heavy Use | *Euclea divinorum* | 55 | 85 | −30 |
| Heavy Use | *Ziziphus mucronata* | 35 | 55 | −20 |
| Moderate Use | *Terminalia sericea* | 70 | 20 | 50 |
| Moderate Use | *Albizia harveyi* | 65 | 18 | 47 |
| Moderate Use | *Grewia flavescens* | 60 | 25 | 35 |
| Moderate Use | *Peltophorum africanum* | 75 | 30 | 45 |

**Table 4.** *Cont.*

| Grazing Zone | Species Name | Recruitment Rate (Seedlings/ha) | Mortality Rate (Plants/ha) | Net Woody Regeneration (Plants/ha) |
|---|---|---|---|---|
| Moderate Use | *Combretum molle* | 68 | 22 | 46 |
| Light Use | *Combretum hereroense* | 90 | 15 | 75 |
| Light Use | *Grewia flava* | 80 | 10 | 70 |
| Light Use | *Brachystegia spiciformis* | 85 | 12 | 73 |
| Light Use | *Sclerocarya birrea* | 95 | 8 | 87 |
| Light Use | *Terminalia prunioides* | 88 | 14 | 74 |
| Riparian Zone | *Vachellia tortilis* | 100 | 5 | 95 |
| Riparian Zone | *Faidherbia albida* | 105 | 7 | 98 |
| Riparian Zone | *Berchemia discolor* | 92 | 6 | 86 |
| Riparian Zone | *Diospyros mespiliformis* | 78 | 10 | 68 |
| Riparian Zone | *Boscia albitrunca* | 110 | 12 | 98 |

*3.7. Soil Characteristics Across Grazing Zones*

Figure 4 illustrates how soil parameters vary across different grazing intensities (heavy, moderate, and light use). Soil pH shows a significant increase with decreasing grazing intensity, with light-use zones having the highest pH (6.8) compared to heavy-use zones (5.8). Organic carbon content similarly increases from heavily grazed areas, with only 1.2%, to lightly grazed zones, which have a higher organic carbon content of 2.4%. Total nitrogen content follows the same trend, being lowest in heavily grazed zones (0.07%) and highest in light-use areas (0.12%). Available phosphorus also shows a marked difference, rising significantly with reduced grazing intensity, from 5.6 mg/kg in heavy-use areas to 12.5 mg/kg in light-use zones. Bulk density, in contrast, decreases with lower grazing pressure; heavily grazed soils exhibit a bulk density of 1.4 g/cm$^3$, while lightly grazed soils are less compact, with a bulk density of 1.1 g/cm$^3$. These trends indicate that lighter grazing promotes improved soil quality, with enhanced nutrient content and reduced soil compaction, which is critical for supporting vegetation health and ecosystem resilience.

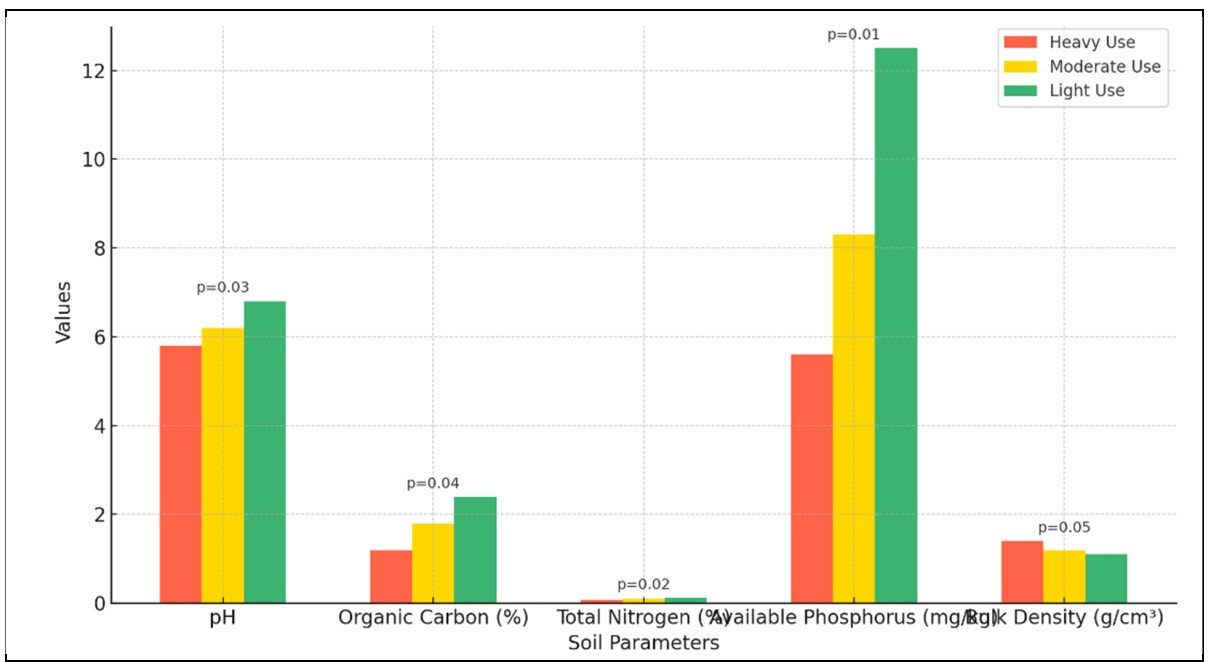

**Figure 4.** Comparison of soil parameters across grazing intensities.

### 3.8. Proximate Nutrient Composition of Fecal Samples from Cattle and Goats

As outlined in Table 5, the proximate nutrient composition of fecal samples from cattle and goats varies significantly across grazing zones. For both species, crude protein content increases from the heavy-use to the light-use zone. Cattle feces from the heavy-use zone have a crude protein content of 12.4%, which is significantly lower than that from the moderate-use (14.2%) and light-use zones (15.1%). Similarly, goats in the heavy-use zone have a crude protein level of 11.8%, significantly lower than that observed in the moderate- (13.5%) and light-use zones (14.7%). Fiber content shows an inverse trend, with cattle and goats in the heavy-use zone exhibiting the highest fiber levels (60.3% and 62.1%, respectively), while the light-use zone has the lowest fiber content (50.7% for cattle and 52.6% for goats), indicating that as grazing intensity decreases, forage quality improves. Digestibility rates are also significantly higher in the light-use zone, with cattle showing 72.8% and goats showing 71.3% digestibility rates, compared to 65.2% and 64.1% in the heavy-use zones, respectively. This suggests that dietary quality and energy availability improve with reduced grazing intensity. Ash content did not show significant differences across grazing zones, with values ranging from 6.1% to 8.0%. However, these variations should be noted, as they can influence nutrient availability and overall forage quality.

**Table 5.** Proximate nutrient composition of fecal samples from cattle and goats (mean $\pm$ SE).

| Species | Grazing Zone | Crude Protein (%) | Fiber Content (%) | Digestibility (%) | Ash (%) |
|---|---|---|---|---|---|
| Cattle | Heavy Use | $12.4 \pm 0.8$ | $60.3 \pm 1.3$ | $65.2 \pm 2.1$ | $7.5 \pm 0.5$ |
| Cattle | Moderate Use | $14.2 \pm 0.7$ | $55.8 \pm 1.5$ | $70.4 \pm 2.3$ | $6.9 \pm 0.4$ |
| Cattle | Light Use | $15.1 \pm 0.6$ | $50.7 \pm 1.2$ | $72.8 \pm 1.8$ | $6.1 \pm 0.3$ |
| Goats | Heavy Use | $11.8 \pm 0.9$ | $62.1 \pm 1.4$ | $64.1 \pm 2.4$ | $8.0 \pm 0.6$ |
| Goats | Moderate Use | $13.5 \pm 0.6$ | $57.2 \pm 1.3$ | $68.7 \pm 2.2$ | $7.2 \pm 0.5$ |
| Goats | Light Use | $14.7 \pm 0.5$ | $52.6 \pm 1.1$ | $71.3 \pm 1.9$ | $6.5 \pm 0.4$ |

### 3.9. Grazing Intensity and Woody Vegetation Characteristics

Table 6 presents the regression analysis results for grazing intensity's effect on woody vegetation. Grazing intensity shows a significant negative association with woody species richness ($\beta = -0.42$, $p = 0.007$, $R^2 = 0.38$), density ($\beta = -1.15$, $p = 0.001$, $R^2 = 0.47$), and crown cover ($\beta = -0.58$, $p = 0.012$, $R^2 = 0.32$), indicating that increased grazing pressure reduces vegetation diversity and density. A marginal effect was found on basal area ($\beta = -0.35$, $p = 0.053$), suggesting that high grazing intensity can limit canopy cover and habitat structure. Additionally, the recruitment rate of woody species is negatively impacted by grazing ($\beta = -0.72$, $p = 0.018$, $R^2 = 0.29$), and the mortality rate of woody plants shows a positive relationship with grazing ($\beta = 0.95$, $p = 0.004$, $R^2 = 0.42$). These findings imply that intense grazing disrupts regeneration and contributes to vegetation degradation.

**Table 6.** Regression analysis of grazing intensity and woody species characteristics.

| Dependent Variable | Independent Variable | Regression Coefficient ($\beta$) | Standard Error | t-Value | *p*-Value | $R^2$ |
|---|---|---|---|---|---|---|
| Woody Species Richness | Grazing Intensity (Heavy to Light) | $-0.42$ | 0.15 | $-2.80$ | 0.007 ** | 0.38 |
| Woody Density (plants/ha) | Grazing Intensity | $-1.15$ | 0.32 | $-3.59$ | 0.001 ** | 0.47 |
| Crown Cover (%) | Grazing Intensity | $-0.58$ | 0.22 | $-2.64$ | 0.012 * | 0.32 |
| Basal Area (cm$^2$) | Grazing Intensity | $-0.35$ | 0.18 | $-1.94$ | 0.053 | 0.25 |
| Recruitment Rate (plants/ha) | Grazing Intensity | $-0.72$ | 0.29 | $-2.48$ | 0.018 * | 0.29 |
| Mortality Rate (plants/ha) | Grazing Intensity | 0.95 | 0.31 | 3.06 | 0.004 ** | 0.42 |

* Significant at $p < 0.05$; ** Significant at $p < 0.0001$.

### 3.10. Livestock Dietary Preferences and Environmental Factors

Table 7 illustrates the multiple regression analysis for cattle and goat diet composition under varying environmental conditions. For cattle, factors like woody cover ($\beta = 0.38$, $p = 0.003$) and grass biomass ($\beta = 0.21$, $p = 0.021$) positively influence diet quality. Seasonal changes also affect diet composition ($\beta = 0.42$, $p = 0.007$). In contrast, soil organic carbon content negatively affects cattle diets ($\beta = -0.18$, $p = 0.027$), suggesting that soil quality impacts forage availability. For goats, woody species density ($\beta = 0.45$, $p = 0.006$) and shrub cover ($\beta = 0.28$, $p = 0.033$) are significant positive predictors of diet composition, whereas grass biomass negatively affects their diet ($\beta = -0.22$, $p = 0.029$), indicating goats' preference for browse-rich areas.

**Table 7.** Multiple regression analysis of factors influencing livestock dietary preferences.

| Dependent Variable | Independent Variables | Regression Coefficient ($\beta$) | Standard Error | t-Value | *p*-Value | $R^2$ |
|---|---|---|---|---|---|---|
| Cattle Diet Composition | Woody Species Cover (%) | 0.38 | 0.12 | 3.17 | 0.003 ** | 0.49 |
| | Grass Biomass (kg/ha) | 0.21 | 0.09 | 2.33 | 0.021 * | |
| | Shrub Height (cm) | 0.10 | 0.05 | 2.00 | 0.049 * | |
| | Seasonality (Dry = 1, Wet = 0) | 0.42 | 0.15 | 2.80 | 0.007 ** | |
| | Soil Organic Carbon (%) | −0.18 | 0.08 | −2.25 | 0.027 * | |
| Goat Diet Composition | Woody Species Density (plants/ha) | 0.45 | 0.16 | 2.81 | 0.006 ** | 0.51 |
| | Shrub Cover (%) | 0.28 | 0.13 | 2.15 | 0.033 * | |
| | Grass Biomass (kg/ha) | −0.22 | 0.10 | −2.20 | 0.029 * | |
| | Basal Area (cm$^2$/plant) | 0.11 | 0.07 | 1.57 | 0.120 | |

* Significant at $p < 0.05$; ** Significant at $p < 0.0001$.

### 3.11. Interrelations Between Vegetation and Livestock Parameters

Table 8 summarizes the correlation matrix for vegetation and livestock parameters. Grazing intensity correlates negatively with woody species richness (r = −0.56), grass biomass (r = −0.62), and crown cover (r = −0.48), showing that heavy grazing reduces vegetation quality and diversity. There is a positive correlation between cattle and goat diets (r = 0.49), suggesting an overlap in dietary preferences in shared grazing areas, potentially leading to competition.

**Table 8.** Correlation matrix of vegetation and livestock parameters.

| | Grazing Intensity | Woody Species Richness | Grass Biomass (kg/ha) | Crown Cover (%) | Cattle Diet (%) | Goat Diet (%) |
|---|---|---|---|---|---|---|
| Grazing Intensity | 1.00 | −0.56 ** | −0.62 ** | −0.48 * | 0.37 * | 0.29 |
| Woody Richness | −0.56 ** | 1.00 | 0.41 * | 0.53 ** | −0.24 | −0.18 |
| Grass Biomass | −0.62 ** | 0.41 * | 1.00 | 0.48 * | −0.15 | −0.22 |
| Crown Cover | −0.48 * | 0.53 ** | 0.48 * | 1.00 | −0.12 | −0.32 * |
| Cattle Diet | 0.37 * | −0.24 | −0.15 | −0.12 | 1.00 | 0.49 ** |
| Goat Diet | 0.29 | −0.18 | −0.22 | −0.32 * | 0.49 ** | 1.00 |

* Significant at $p < 0.05$; ** Significant at $p < 0.0001$.

### 3.12. Predictive Analysis of Livestock Productivity

Table 9 shows the stepwise regression results for predicting livestock productivity based on vegetation and soil characteristics. Grass biomass emerged as the most critical

factor for livestock productivity (β = 0.62, $p < 0.01$, $R^2 = 0.39$), followed by woody species richness, which added explanatory power to the model (β = 0.48, $p < 0.01$).

**Table 9.** Stepwise regression analysis for predicting livestock productivity based on vegetation and soil characteristics.

| Step | Independent Variable | β | $R^2$ Change | Model $R^2$ | *p*-Value |
|:---:|:---:|:---:|:---:|:---:|:---:|
| 1 | Grass Biomass (kg/ha) | 0.62 ** | 0.39 | 0.39 | 0.001 ** |
| 2 | Woody Species Richness | 0.48 ** | 0.24 | 0.63 | 0.003 ** |
| 3 | Soil Organic Carbon (%) | 0.36 * | 0.15 | 0.78 | 0.017 * |
| 4 | Basal Area (cm$^2$/plant) | 0.33 * | 0.09 | 0.87 | 0.045 * |

* Significant at $p < 0.05$; ** Significant at $p < 0.0001$.

## 4. Discussion

### 4.1. Influence of Grazing Intensity on Herbaceous and Woody Species Composition

Grazing intensity significantly influenced the composition and abundance of herbaceous and woody species across the grazing zones. Heavily grazed areas exhibited a reduction in herbaceous species alongside an increase in less palatable woody species like *Dichrostachys cinerea* and *Senegalia mellifera*, corroborating findings from [4,21]. This shift toward woody dominance under heavy grazing reduces high-quality forage availability, impacting rangeland productivity. Moderately grazed areas displayed a mix of herbaceous and woody species, supporting greater species diversity and aligning with the intermediate disturbance hypothesis [17]. Lightly grazed zones, with the highest species richness and palatable vegetation, promote forage sustainability and ecosystem stability, as supported by [16]. Meanwhile, riparian areas hosted abundant deep-rooted woody species like *Vachellia tortilis*, which aid in soil stabilization but may limit herbaceous cover and access to water.

### 4.2. Grazing Effects on Forage Nutritional Composition

Nutritional composition varied with grazing intensity, as heavily grazed zones showed lower crude protein (CP) content and digestibility. This result is consistent with [4,10], who noted reduced forage quality under heavy grazing. Forage quality improved in moderately grazed zones, where nutrient-rich species like *Panicum maximum* thrived. Lightly grazed zones demonstrated the highest CP content and digestibility, linked to the presence of high-quality species such as *Sclerocarya birrea*, reinforcing findings from [2]. Riparian zones offered high CP forage, supported by nitrogen-fixing species like *Faidherbia albida*, yet required managed woody density to optimize forage access.

### 4.3. Seasonal Biomass Dynamics and Grazing Pressure

Seasonal fluctuations in herbaceous biomass and cover were prominent, with wet-season growth enhancing forage availability, especially in lightly grazed and riparian areas [10,36]. Heavily grazed zones, however, showed significant dry-season declines due to the compounded effects of grazing and moisture stress, leading to reduced forage for livestock [37]. In contrast, moderately and lightly grazed zones retained higher biomass during the dry season, supported by drought-resistant species like *Cenchrus ciliaris* [38]. Riparian zones, with access to groundwater, exhibited resilience to seasonal droughts, serving as critical dry-season refuges [39]. Effective management of grazing in riparian and heavily grazed zones can prevent vegetation loss, sustaining rangeland resilience.

### 4.4. Woody Species Characteristics and Utilization by Cattle and Goats

The composition and characteristics of woody species varied significantly across grazing zones, influencing their utilization by cattle and goats. Heavily grazed zones exhibited lower woody species richness and density compared to moderately and lightly grazed areas.



This pattern can be attributed to intense grazing pressure and trampling, which impede the recruitment and growth of woody seedlings and increase vulnerability to herbivory [5,10]. Heavy grazing not only reduces species richness but also alters the structural attributes of woody vegetation, such as basal area and canopy cover, which can have cascading effects on forage availability and habitat quality for both livestock and wildlife [40]. In contrast, lightly grazed areas had significantly higher woody species richness and density, likely due to reduced browsing pressure and less frequent disturbances. The presence of diverse and dense woody vegetation in lightly grazed zones suggests that lower grazing intensities allow for better recruitment and survival of woody species, promoting a balanced coexistence of grasses and shrubs [41]. This finding aligns with the grazing optimization theory, which posits that moderate to light grazing can enhance plant community diversity by creating spatial heterogeneity in plant resources, thereby preventing the dominance of a few competitive species [42].

Woody species such as *Acacia karroo* and *Dichrostachys cinerea* were more abundant in heavily grazed zones, indicating their resilience and ability to thrive under high grazing pressure. This finding is consistent with other studies which identify these species as indicators of rangeland degradation due to their ability to proliferate in overgrazed environments where the herbaceous layer is depleted [4,10]. The encroachment of such woody species can significantly alter rangeland dynamics, as their dense growth forms reduce grass cover and compete for resources, leading to decreased forage availability for grazers like cattle [39]. The dominance of *Dichrostachys cinerea*, for instance, has been reported to impede grass regrowth and reduce overall pasture productivity, thereby impacting livestock-carrying capacity (Samuels et al., 2016) [12].

The utilization of woody species by cattle and goats also reveals distinct patterns, reflecting their different dietary preferences and foraging behaviors. Cattle, being predominantly grazers, primarily utilized woody vegetation during the dry season when herbaceous forage was scarce. Their preference for woody species with palatable leaves and accessible branches, such as *Terminalia sericea* and *Combretum molle*, underscores the role of these plants as supplementary forage resources during periods of forage scarcity [16]. According to [21], such species provide critical nutrients, especially protein and minerals, which help maintain cattle body condition and reduce weight loss during dry spells. Goats, on the other hand, are mixed feeders with a strong preference for browse, showing a higher utilization of woody species across all seasons. They consume a wider range of woody species, including less palatable ones like *Senegalia mellifera* and *Euclea divinorum*, due to their ability to selectively browse on leaves, twigs, and even bark [22]. This adaptability allows goats to exploit a broader range of forage resources and maintain productivity under varying environmental conditions [4]. The higher utilization of woody species by goats, especially in heavily grazed zones, underscores their competitive advantage in degraded rangelands where herbaceous forage is limited [20].

Moreover, the results indicate that the characteristics of woody species, such as canopy cover, basal area, and shrub height, significantly influence livestock dietary preferences. Cattle tend to utilize woody species with larger basal areas and lower canopy heights, as these features allow easier access to foliage and browsing material [10]. However, goats prefer species with dense canopies and taller heights, which provide a greater variety of browsing options, including leaves, pods, and fruits [43]. This finding is supported by the work of [39], who noted that goats' ability to reach higher browse material and selectively feed on specific plant parts gives them a foraging advantage in areas with dense woody encroachment. The impact of grazing intensity on woody species utilization is also evident. Increased browsing pressure in heavily grazed zones leads to significant changes in woody plant structure and regeneration dynamics. High browsing intensity can reduce shoot and leaf biomass, affecting the reproductive capacity of woody plants and their ability to withstand environmental stress [40]. This, in turn, can lead to shifts in species composition, favoring resilient but less palatable species such as *Ziziphus mucronata*, which further reduces the forage quality available to livestock [13].

### 4.5. Livestock Diet Overlap and Resource Competition Index

The analysis of livestock diet overlap and the resource competition index reveals important insights into interspecific resource utilization and the potential for competition between cattle and goats in different grazing zones. The degree of dietary overlap between cattle and goats varied significantly across grazing intensities, reflecting differences in their foraging strategies and resource availability in each zone. In heavily grazed areas, dietary overlap was highest, suggesting a high degree of competition for limited forage resources. This observation aligns with the findings of [44], who reported that when forage availability is low, competition between herbivores intensifies, potentially leading to reduced intake and body condition for one or both species. The high dietary overlap in heavily grazed zones indicates that both cattle and goats are forced to consume similar forage species, such as *Acacia karroo* and *Dichrostachys cinerea*, which are known to proliferate in degraded areas [45]. This overlap is likely driven by the scarcity of preferred herbaceous species and the encroachment of less palatable woody species, which become the primary available forage. This pattern can be detrimental to cattle, which primarily rely on grasses for their diet, as they may be outcompeted by goats, which are better adapted to browsing [17]. Goats, being more flexible feeders, can shift their diet more readily to include a higher proportion of woody species and browse, thus gaining a competitive advantage in environments where grass availability is low [10].

In contrast, moderate and lightly grazed zones exhibited lower levels of diet overlap between the two species. In these areas, the greater availability and diversity of forage resources, including a balanced mixture of grasses and woody species, reduce direct competition for the same plant species. This finding aligns with the niche differentiation theory, which posits that resource partitioning in more heterogeneous environments allows coexisting species to exploit different forage resources, thereby minimizing direct competition [46]. For example, in lightly grazed zones, cattle predominantly graze on perennial grasses such as *Themeda triandra* and *Digitaria eriantha*, which are abundant and provide high nutritional quality [9]. Meanwhile, goats focus on a mix of grasses and palatable shrubs such as *Grewia flava* and *Boscia albitrunca*, reducing the likelihood of significant dietary overlap [3]. The resource competition index further supports these findings by quantifying the intensity of competition between the two species across grazing zones. The index values were highest in heavily grazed areas, indicating severe competition for limited forage. High competition indices in degraded rangelands have been documented in several studies, including in work by [39], who found that heavy grazing pressure leads to the overutilization of key forage species, resulting in lower dietary intake and poorer animal performance. The implications of high competition are particularly concerning for cattle, which are less adaptable to shifts in forage type compared to goats. Cattle often experience reduced weight gain and reproductive performance under conditions of high dietary competition [7].

In moderately grazed zones, the competition index was significantly lower, indicating that both species were able to utilize different forage resources with minimal overlap. This finding aligns with studies by [40], which observed that moderate grazing promotes a more diverse plant community, thereby supporting niche partitioning among herbivores. The presence of a diverse herbaceous layer, combined with sufficient woody cover, provides complementary forage resources that meet the dietary needs of both grazers and browsers without intense competition [47]. The lowest competition index was recorded in lightly grazed zones, where resource availability is highest and dietary overlap is minimal. This finding underscores the importance of maintaining moderate to low grazing intensities to promote forage heterogeneity and reduce interspecific competition [5]. Sufficient grass biomass and a diverse array of woody species enable cattle and goats to select forage that aligns closely with their feeding preferences. For instance, goats can utilize taller shrubs and woody species for browsing, while cattle have access to an abundance of high-quality grasses, resulting in reduced competition and improved livestock productivity [10]. These results are critical for rangeland management, particularly in areas where multiple livestock species graze simultaneously. Managing grazing intensity to maintain a balance between

herbaceous and woody vegetation can reduce resource competition and enhance the productivity of both cattle and goats. Strategies such as rotational grazing and controlled browsing can help mitigate the negative impacts of overgrazing and promote the sustainable use of rangeland resources [39]. By ensuring that grazing pressure is kept within sustainable limits, rangeland managers can support higher forage availability and quality, thereby minimizing competition and promoting the coexistence of different livestock species [48].

*4.6. Seasonal Changes in Woody Species Recruitment and Mortality Rates*

The results indicate significant seasonal variations in woody species recruitment and mortality rates across different grazing zones, likely influenced by varying grazing pressure, climatic conditions, and species-specific resilience. The heavy-use zones displayed negative net woody regeneration for all assessed species, suggesting that recruitment is unable to keep pace with high mortality rates. This trend is consistent with the findings of [10], who reported that high grazing pressure often leads to elevated mortality in young woody plants due to trampling and browsing by livestock, compounded by competition with herbaceous species for limited soil moisture and nutrients. In particular, the species *Acacia karroo* and *Dichrostachys cinerea* exhibited higher mortality rates compared to their recruitment in heavily grazed zones, resulting in a net loss of 25 and 30 plants/ha, respectively. These results reflect the susceptibility of these species to intensive browsing, which disrupts their growth and regeneration patterns [13]. Intensive grazing pressure often reduces the capacity of woody species to establish and survive, as younger saplings are repeatedly browsed or damaged by livestock. Additionally, heavy grazing tends to alter soil structure and reduce seedling establishment, further contributing to the observed negative net regeneration [16]. Conversely, in moderate- and light-use zones, a more balanced relationship between recruitment and mortality was observed, with positive net woody regeneration values across most species. This positive net regeneration in moderate-use zones, for example, for species like *Terminalia sericea* (+50 plants/ha) and *Combretum molle* (+46 plants/ha), suggests that reduced grazing pressure allows for better recruitment and survival rates. This pattern aligns with the findings of [2], who noted that moderate grazing can promote a diverse plant community by reducing the dominance of certain species and allowing others to regenerate successfully. The presence of sufficient soil cover and lower trampling intensity in these zones likely provides a more conducive environment for seedling establishment and growth, supporting higher regeneration rates.

Interestingly, the light-use zones showed the highest recruitment rates across all assessed species, with *Sclerocarya birrea* and *Terminalia prunioides* achieving net gains of +87 and +74 plants/ha, respectively. These findings indicate that minimal disturbance fosters the best conditions for woody species regeneration. Studies by [14,39] support this observation, suggesting that light grazing reduces the intensity of competitive interactions and physical damage to young plants, thereby enhancing their survival and growth potential. Furthermore, light-grazing zones typically experience less soil compaction, which improves water infiltration and nutrient availability—key factors for successful woody species recruitment [36]. The riparian zones exhibited the highest net woody regeneration values, particularly for species such as *Vachellia tortilis* (+95 plants/ha) and *Faidherbia albida* (+98 plants/ha). This trend can be attributed to the more favorable microclimatic and soil moisture conditions typically found in riparian areas, which enhance seedling establishment and survival compared to drier upland areas [49]. The high regeneration in these areas, despite being accessible to livestock, may also be due to the resilience of these riparian species, which are adapted to periodic disturbances such as flooding and grazing. As a result, riparian zones act as refuges for woody species, allowing them to maintain viable populations even under moderate grazing pressures [10]. However, the observed patterns of recruitment and mortality suggest that without proper management, heavy-use zones may continue to experience a decline in woody species density and diversity, potentially leading to shifts in vegetation structure and function over time. Negative net woody

regeneration values in these zones could contribute to long-term woody species loss and an increase in bare ground, exacerbating soil erosion and land degradation [4]. Therefore, targeted management interventions, such as the rotational grazing or seasonal resting of heavily grazed areas, are essential to reduce pressure on young plants and promote their successful establishment [39].

*4.7. Soil Characteristics Across Grazing Zones*

The analysis of soil characteristics across different grazing zones revealed distinct variations in key properties such as pH, organic carbon content, total nitrogen, available phosphorus, and bulk density. These variations demonstrate the impact of grazing intensity on soil health and fertility and align with findings from similar studies in rangeland ecosystems, which indicate that grazing pressure affects soil nutrient dynamics, physical structure, and overall soil quality [17,50]. Specifically, soil pH values increased progressively from heavy-use zones (pH 5.8) to light-use zones (pH 6.8), with statistically significant differences ($p = 0.03$). The lower pH in heavily grazed areas can be attributed to high grazing intensity, which compacts the soil, reduces infiltration, and increases surface runoff, thereby promoting the leaching of base cations such as calcium and magnesium [10]. As grazing intensity decreases, reduced soil compaction and increased vegetative cover help mitigate leaching, stabilize pH levels, and create more favorable conditions for plant growth [51].

Organic carbon content exhibited a clear gradient, ranging from 1.2% in heavy-use zones to 2.4% in light-use zones ($p = 0.04$), indicating that grazing intensity significantly influences soil organic matter (SOM) accumulation. Heavy grazing often leads to the removal of plant biomass, reduced root density, and less organic matter input into the soil [40]. Consequently, soil organic carbon tends to be lower in heavily grazed areas due to the diminished contribution of leaf litter and root exudates, which are key sources of organic carbon [23]. In contrast, light-use zones experience reduced disturbance and higher vegetative cover, facilitating greater accumulation of organic residues and enhancing SOM content. This pattern is consistent with findings from [52], who reported that moderate to light grazing can improve soil carbon content by promoting plant growth and organic matter inputs. Total nitrogen content showed a similar trend, with significantly lower values in heavy-use zones (0.07%) compared to light-use zones (0.12%) ($p = 0.02$). The reduction in nitrogen levels in heavily grazed areas can be linked to increased soil erosion, lower soil organic matter (SOM) content, and reduced nitrogen fixation by leguminous species due to overgrazing [4]. Legumes, which contribute nitrogen to the soil through biological nitrogen fixation, are particularly sensitive to heavy grazing pressure. Their decline in heavily grazed areas results in lower nitrogen availability in the soil [21]. In contrast, moderate- and light-use zones maintain a higher proportion of legumes and vegetative cover, which improves nitrogen retention and cycling within the soil system [10]).

Phosphorus availability also varied significantly across grazing zones, with the lowest concentration recorded in heavily grazed areas (5.6 mg/kg) and the highest in light-use areas (12.5 mg/kg) ($p = 0.01$). Phosphorus availability is strongly influenced by organic matter content and soil microbial activity, both of which are typically reduced under high grazing pressure [15]. Heavy grazing can disrupt the phosphorus cycle by compacting the soil, reducing microbial activity, and limiting the decomposition of organic matter, leading to lower phosphorus availability [14]. In contrast, light grazing promotes microbial activity and organic matter decomposition, enhancing phosphorus mineralization and availability to plants [5]. Bulk density values were significantly higher in heavy-use zones (1.4 g/cm$^3$) compared to light-use zones (1.1 g/cm$^3$) ($p = 0.05$), indicating increased soil compaction in heavily grazed areas. Soil compaction, often associated with intensive grazing, reduces pore space, limits water infiltration, and restricts root growth [39]. These effects can have a cascading impact on plant growth and soil nutrient availability, ultimately leading to reduced soil productivity and vegetation cover [4]. In contrast, light grazing minimizes compaction and maintains soil structure, thereby improving bulk density and supporting a more stable and productive soil environment [11].

*4.8. Proximate Nutrient Composition of Fecal Samples from Cattle and Goats*

The analysis of the proximate nutrient composition of fecal samples from cattle and goats across varying grazing zones revealed distinct differences in crude protein, fiber content, digestibility, and ash percentages. These findings highlight the influence of diet quality and grazing intensity on nutrient intake and digestion. They are consistent with similar studies on livestock nutrition, which demonstrate that fecal nutrient content serves as a reliable indicator of forage quality and animal nutritional status [12,41]. Crude protein (CP) content in fecal samples was notably higher in light-use grazing zones for both cattle (15.1 ± 0.6%) and goats (14.7 ± 0.5%) compared to the lower values observed in heavy-use zones (12.4 ± 0.8% for cattle and 11.8 ± 0.9% for goats). The reduced CP content in heavily grazed areas suggests that animals are consuming lower-quality forage with diminished protein availability, likely due to overgrazing and the resultant decline in high-quality, protein-rich forage species [6,10]. Overgrazing can deplete preferred forages such as legumes, leading to a shift in diet composition toward grasses and woody plants that may have lower protein content, thereby reducing overall protein intake [53]. In contrast, light grazing maintains a higher abundance of protein-rich species, allowing animals to select higher-quality forage, which is reflected in the increased fecal protein content [18].

Fiber content, as measured by neutral detergent fiber (NDF), was highest in fecal samples from heavy-use zones for both cattle (60.3 ± 1.3%) and goats (62.1 ± 1.4%). This suggests that animals in these areas are consuming more mature and fibrous plant material, which has lower digestibility and energy content [10]. High fiber content in feces is often an indicator of low forage quality, as increased fiber is associated with lower nutrient availability and reduced intake of digestible nutrients [17]. These findings align with other studies which show a strong positive correlation between grazing pressure and dietary fiber, resulting from the depletion of preferred forage species and a subsequent reliance on less palatable, more fibrous vegetation [20]. The lower fiber content observed in light-use zones (50.7 ± 1.2% for cattle and 52.6 ± 1.1% for goats) indicates higher-quality forage with a lower proportion of mature stems and higher leaf-to-stem ratios, thereby enhancing nutrient digestibility [22]. Digestibility values followed a similar trend, being significantly higher in light-use zones (72.8 ± 1.8% for cattle and 71.3 ± 1.9% for goats) compared to heavy-use zones (65.2 ± 2.1% for cattle and 64.1 ± 2.4% for goats). This variation in digestibility is linked to forage quality, as light-grazing zones typically contain a greater abundance of green, leafy material that is easier to break down and assimilate [50]. Digestibility is a key determinant of energy intake, and the lower digestibility rates observed in heavily grazed zones can negatively impact animal performance due to reduced energy extraction from consumed forage, ultimately leading to decreased weight gain and milk production [49]. The results corroborate findings from other rangeland studies, which report that digestibility declines under heavy grazing due to the increased consumption of woody and fibrous materials [45].

Ash content was also highest in the feces of animals grazing in heavy-use zones (7.5 ± 0.5% for cattle and 8.0 ± 0.6% for goats) compared to light-use zones (6.1 ± 0.3% for cattle and 6.5 ± 0.4% for goats). High ash content in feces can indicate a greater intake of soil and low-quality plant material, often associated with overgrazed pastures where nutrient-rich forage is scarce [10]. Animals in degraded areas may ingest more soil while grazing close to the ground or consume plants with high mineral content, resulting in elevated fecal ash levels [38]. This trend is supported by [15], who found that increased grazing pressure led to higher ash content in feces due to soil contamination and lower forage quality. Interestingly, goats consistently exhibited higher fiber and ash content compared to cattle across all grazing zones, while cattle showed higher crude protein and digestibility values. These differences are likely due to species-specific foraging strategies and digestive adaptations [10]. Goats are known to be more selective feeders, often consuming a greater proportion of woody plants and shrubs that have a higher fiber and ash content than the grasses preferred by cattle [4]. Additionally, the rumen morphology and microbial

populations of cattle are better adapted to breaking down fibrous material, which could explain their higher digestibility and protein utilization compared to goats [36].

### 4.9. Rangeland Degradation and Restoration Strategies

Rangeland degradation remains a pressing challenge, primarily driven by overgrazing, poor land management, and climate variability. These factors contribute to vegetation loss, soil erosion, and reduced productivity, negatively affecting livestock and livelihoods [54,55]. The findings highlight that grazing management practices, such as rotational grazing and controlled grazing zones, can mitigate degradation by promoting balanced plant recovery and soil stability, aligning with principles outlined by [27,32]. Restoration efforts, particularly in heavily degraded areas, should focus on reseeding with native species and reducing grazing intensity to facilitate ecosystem recovery and sustain biodiversity. Effective restoration can help reverse degradation, improve forage availability, and support livestock productivity in the long term [10,56].

### 4.10. Implications for Livestock Management and Future Rangeland Sustainability

The impact of grazing on forage quality and ecosystem resilience has crucial implications for livestock management. As observed, heavy grazing reduces forage nutritional quality, demanding supplemental feeding strategies to meet livestock dietary needs [15]. By contrast, moderate grazing supports higher forage quality and biodiversity, which are beneficial for livestock growth and ecosystem health [17]. Seasonal variations further underscore the need for adaptive management; maintaining livestock productivity requires access to critical grazing refuges, such as riparian zones, during dry periods to offset forage scarcity [25]. The integration of these findings suggests that sustainable rangeland management should prioritize adaptive grazing strategies, restoration, and the selective thinning of woody species to balance forage quality and accessibility. Future strategies could benefit from continued monitoring of species composition and resilience indicators, aiding the development of region-specific approaches for sustainable livestock systems in semi-arid rangelands [42,51].

### 4.11. Study Limitations

This study has several notable limitations. First, it employed a cross-sectional design, capturing only a snapshot of the interactions between vegetation and livestock at a specific point in time. Consequently, it fails to account for seasonal and annual variations in vegetation dynamics and livestock productivity. This limitation restricts our understanding of how these interactions evolve over longer periods, which is crucial for effective rangeland management. Second, this study primarily focused on biophysical factors influencing livestock productivity, neglecting the potential impact of socio-economic factors such as herd management practices, breed differences, and market conditions. These variables can significantly affect livestock performance and are essential for a more comprehensive understanding of productivity drivers. Additionally, the reliance on fecal analysis to infer dietary preferences may introduce inaccuracies, as this method may not fully capture complete dietary intake, particularly when distinguishing between similar forage types. This study also examined only two livestock species, cattle and goats, limiting its applicability to other grazers or browsers in similar ecosystems. Finally, this research did not consider the potential interactions between domestic livestock and wildlife, which could significantly influence grazing dynamics and resource competition. This omission may lead to an incomplete understanding of how grazing intensity impacts rangeland vegetation and livestock productivity.

### 4.12. Future Research Directions

Future research should focus on longitudinal studies to capture seasonal and annual variations in vegetation and livestock productivity, thereby providing a deeper understanding of temporal changes and their implications. Expanding the study to include additional

livestock species and mixed-species grazing systems would offer a more comprehensive view of the impacts of grazing on vegetation dynamics. Furthermore, integrating remote sensing and GIS technologies could enhance monitoring capabilities by providing high-resolution data on vegetation cover and changes over time, allowing for more precise assessments of rangeland conditions and their relationship with livestock performance. Incorporating socio-economic factors, such as herder practices and market dynamics, would further strengthen future studies, offering a more complete perspective on the influences affecting livestock productivity. Lastly, research should also examine the impacts of climate change on vegetation patterns and livestock productivity to identify adaptive management strategies that can support rangeland sustainability in the face of changing environmental conditions.

## 5. Conclusions

This study offers a thorough assessment of the interactions among grazing intensity, vegetation dynamics, and livestock productivity in arid and semi-arid rangelands. The findings emphasize the intricate relationships between grazing pressure and the characteristics of both herbaceous and woody vegetation, underscoring the vital importance of sustainable grazing management for rangeland health. Increased grazing intensity was linked to a decline in woody species richness, a reduction in grass biomass, and changes in vegetation structure, which subsequently affected the dietary preferences and productivity of livestock species. The results showed that cattle and goats respond differently to variations in vegetation availability and quality, with notable overlap in their diets in heavily grazed areas, leading to heightened competition for resources. Furthermore, this study indicated that grazing pressure not only affects vegetation composition and cover but also influences essential soil parameters, such as organic carbon and nutrient availability, thereby connecting vegetation and soil health to livestock productivity outcomes. Regression and path analyses identified grass biomass, woody species richness, and soil characteristics as key predictors of livestock productivity. This finding suggests that managing grazing pressure to optimize these variables could enhance both rangeland productivity and sustainability. However, this study also highlighted the necessity for a more integrated approach that considers both biophysical and socio-economic factors in rangeland management.

**Author Contributions:** Conceptualization: M.S. and I.F.J.; data curation: M.S. and I.F.J.; analysis: M.S.; visualization: M.S.; writing the original draft: M.S.; manuscript editing: I.F.J. All authors have read and agreed to the published version of the manuscript.

**Funding:** The authors acknowledge the financial support provided by the National Research Foundation, grant number TS64 (UID: 99787).

**Data Availability Statement:** Data will be available upon reasonable request.

**Acknowledgments:** The authors express their gratitude to colleagues from the Centre for Global Change (CGC) and the Department of Livestock and Pasture Science at the University of Fort Hare for their valuable feedback and assistance in the development of this manuscript.

**Conflicts of Interest:** The authors declare that there are no commercial or financial relationships that could be perceived as potential conflicts of interest in this research.

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
