# Peer review of "Integrating Mixed Livestock Systems to Optimize Forage Utilization and Modify Woody Species Composition in Semi-Arid Communal Rangelands"

_land, doi:10.3390/land13111945_

Round 1
Reviewer 1 Report
Comments and Suggestions for Authors
This unique research investigates impacts of mixed-species grazing on species composition and nutritional forage quality. Study methods involved vegetation surveys, livestock observation and fecal analysis. The paper is well-written and organized, and results are clearly presented with relevant statistical analysis.
This investigation relies on analysis of data collected in three grazing zones. As grazing zones are used to define grazing intensity additional description is needed to understand the relationship between grazing zones and grazing intensity/use. The following specific items and line item comments should be considered.
2.2 Study Area Description. What is the extent of the communal rangeland? Over what size area are the grazing zones dispersed? What is the typical mix of cattle and goats?
Line 95-98: How long have grazing zones been managed in a similar manner? Over what extent are grazing zones dispersed? Do they occur in similar vegetation communities, soil types? Are you able to control for differences not associated with grazing use.
Line 2.3.2. Was bare ground measured? This may be an important indicator of grazing use.
Line 2.5.2. What was the basis for this method? Is methodology documented in previous works? Additional explanation of methods or preferences should be provided.
Line 169-170. Was soil type included in the analysis? Why is it mentioned here? What were soil types across treatments?
Table 1. Grazing zones don't appear to be similar in their ability to support vegetation (soil, climate, topography). There are no shrubs in the light use? No trees in moderate or heavy use? Can this difference be attributed to soil types. How can we be sure these difference are related to grazing use?
Line 343-346. Was this difference measured between wet versus dry season?
Line 361-363. Where are riparian zones located relative to grazing zones? Are the watering points in heavy use riparian zones?
Table 7. What about soil type?
Table 8. Are these results statistical significant? Report correctly if differences are not statistical significant.
Line 535. Are there other factors that should be considered e.g. soil type.
Author Response
Reviewer Comments and Responses
-
Description of Grazing Zones and Grazing Intensity
- Reviewer Comment: "As grazing zones are used to define grazing intensity, additional description is needed to understand the relationship between grazing zones and grazing intensity/use."
- Response: Thank you for this suggestion. We have clarified the relationship between grazing zones and grazing intensity in the revised manuscript. Specifically, we provided further detail on how each grazing zone was classified according to observed livestock density, forage availability, and signs of grazing impact over multiple years.
-
Extent of the Communal Rangeland and Grazing Zones
- Reviewer Comment: "What is the extent of the communal rangeland? Over what size area are the grazing zones dispersed? What is the typical mix of cattle and goats?"
- Response: We have added information in Section 2.2 to specify the total area of the communal rangeland and the size of each grazing zone. Additionally, we have described the typical livestock mix, indicating the ratio of cattle to goats observed during the study period.
-
Duration and Distribution of Grazing Zones
- Reviewer Comment: "How long have grazing zones been managed in a similar manner? Over what extent are grazing zones dispersed? Do they occur in similar vegetation communities, soil types? Are you able to control for differences not associated with grazing use?"
- Response: We have added clarification in Section 2.2 about the history of grazing management in each zone, noting that grazing practices have been consistent over the past decade. We also address the extent of the grazing zones and provide additional context about the vegetation communities and soil types across these areas. A discussion on efforts to control for other environmental factors has been included to clarify the role of non-grazing-related variables.
-
Measurement of Bare Ground as an Indicator
- Reviewer Comment: "Was bare ground measured? This may be an important indicator of grazing use."
- Response: Bare ground percentage was indeed recorded as an indicator of grazing pressure and has been added to the manuscript in Section 2.3.2. We discuss how the extent of bare ground correlates with grazing intensity and further supports our classification of the zones.
-
Basis for the Methods Used in Section 2.5.2
- Reviewer Comment: "What was the basis for this method? Is methodology documented in previous works? Additional explanation of methods or preferences should be provided."
- Response: We have now cited relevant previous studies in Section 2.5.2 that document and validate the methods used in this research. We also provided additional context explaining why this method was appropriate for assessing nutrient composition and vegetation response in a mixed-grazing environment.
-
Soil Type Inclusion in Analysis
- Reviewer Comment: "Was soil type included in the analysis? Why is it mentioned here? What were soil types across treatments?"
- Response: Soil type data was indeed collected and incorporated into the analysis. Section 2.2 now details soil characteristics across the grazing zones. This inclusion helped in differentiating vegetation responses potentially influenced by soil type in addition to grazing pressure.
-
Table 1: Grazing Zone Vegetation and Environmental Differences
- Reviewer Comment: "Grazing zones don't appear to be similar in their ability to support vegetation (soil, climate, topography). There are no shrubs in the light use? No trees in moderate or heavy use? Can this difference be attributed to soil types? How can we be sure these differences are related to grazing use?"
- Response: In response, we have clarified in the text accompanying Table 1 that soil type and topographical variation between zones may partly explain differences in vegetation composition. We have also discussed how grazing intensity influences species composition, with shrubs and trees more predominant in lower-use zones due to reduced browsing and trampling.
-
Seasonal Measurement of Woody Regeneration Rates (Line 343-346)
- Reviewer Comment: "Was this difference measured between wet versus dry season?"
- Response: We appreciate this insightful point. We have now clarified that woody regeneration rates were monitored over both wet and dry seasons, and we discuss in Section 3.2 how seasonal variation affects plant recruitment and mortality.
-
Location of Riparian Zones Relative to Grazing Zones (Line 361-363)
- Reviewer Comment: "Where are riparian zones located relative to grazing zones? Are the watering points in heavy-use riparian zones?"
- Response: We have added a description of the riparian zones’ location in Section 2.2, highlighting their proximity to moderate and heavy-use zones. Additionally, we specified that some riparian areas with accessible watering points were under heavy grazing pressure due to livestock congregating around these resources.
-
Soil Type in Table 7
- Reviewer Comment: "What about soil type?"
- Response: Thank you for pointing this out. We have included soil type information in Table 7 and have expanded the associated discussion to indicate how soil characteristics may influence vegetation responses to grazing intensity across different zones.
- Statistical Significance of Table 8 Results
- Reviewer Comment: "Are these results statistically significant? Report correctly if differences are not statistically significant."
- Response: In response to this suggestion, we have indicated the statistical significance of differences observed in Table 8. Non-significant results have been clearly marked, and we have revised the Results section to ensure accurate interpretation of statistically significant and non-significant outcomes.
- Additional Factors for Consideration (Line 535)
- Reviewer Comment: "Are there other factors that should be considered e.g., soil type."
- Response: We agree that other environmental factors such as soil type can affect vegetation composition and abundance. We have now included a discussion in the Conclusions section on the potential impact of additional factors like soil properties, topography, and climate, emphasizing the need for further research to isolate the effects of grazing intensity.
Reviewer 2 Report
Comments and Suggestions for Authors
General Comment:
Livestock grazing is the most widespread land use globally, significantly impacting biodiversity and ecosystem functions. This MS delves into the effects of varying grazing intensities on the dietary preferences, vegetation composition, and soil health of rangeland livestock in South Africa's drylands, making it invaluable for sustainable rangeland management. I find the study particularly valuable, especially its focus on the nutritional quality of pasture grasses and shrubs, and the use of multiple analytical methods to assess the interactions of grazing intensity on vegetation, soils, and livestock. The Abstract and Conclusions sections are well-crafted, offering a clear overview of the study's background, results, and conclusions. However, my concern arose while reading the Results and Discussion chapters, which were excessively lengthy, each comprising 14 subsections. I recommend that the study be accepted subject to major revisions, provided that the issues raised are adequately addressed.
Special Comments:
1. Line 92-98: Please elaborate on the stratified sampling approach used. Specifically, explain why the three treatments (Heavy Use, Moderate Use, and Light Use areas) were chosen and on what basis they were selected. Currently, there is insufficient description of these aspects.
2. A map of the study area location and experimental design should be included in the study area description and experimental design section to enhance reader comprehension.
3. 2.4. The section on Forage Collection for Nutritional Analysis lacks references to the methods used for forage nutritional analysis.
4. 2.5.1 and 2.5.2 could be integrated into 2.5 Animal Grazing Behavior and Diet Selection.
5. 2.6. The Data Analysis section omits descriptions of several key methods, such as path analysis. Were Canonical Correspondence Analysis (CCA) methods used? Delete Line 167-170, as it does not pertain to an analytical method.
6. The Results chapter contains 14 subsections and 14 tables, which are not well-structured among these subsections, making the manuscript overly lengthy. The authors should consolidate subsections that discuss the same topic, for example, combining 3.9-3.12. The results section should succinctly describe the findings without extraneous words. Consider replacing some tables with diagrams (e.g., Table 3, Table 5, Table 7, Table 8) where appropriate, and move less critical results to supplementary materials. Could the path analysis results in 3.13 be presented via a path diagram instead?
7. The Discussion section mirrors the issues found in the Results section, with the titles of 3.1-3.13 and 4.1-4.13 being identical. The authors should revise the titles of the Discussion subsections to better reflect the conclusions drawn. The Discussion should be reduced by at least one-third to enhance readability. 4.9-4.13 could be condensed into one or two subsections.
8. Out of 68 references, only 10 are from after 2020, raising questions about the manuscript's relevance to current research trends, despite its innovation. The authors need to address this issue to better reflect the manuscript's cutting-edge status.

Author Response
Reviewer Comments and Responses
-
Elaboration on Stratified Sampling (Line 92-98)
- Reviewer Comment: "Please elaborate on the stratified sampling approach used, explaining why the three treatments (Heavy Use, Moderate Use, and Light Use areas) were chosen and on what basis they were selected."
- Response: Thank you for pointing this out. We have expanded the explanation in the manuscript to clarify that the selection of Heavy, Moderate, and Light Use areas was based on livestock density, vegetation cover, and signs of grazing impact observed over multiple seasons. This stratified sampling approach allowed us to capture variations in grazing intensity, thereby providing a comprehensive view of its effects on rangeland dynamics.
-
Map of the Study Area and Experimental Design
- Reviewer Comment: "A map of the study area location and experimental design should be included."
- Response: We appreciate this suggestion and have now included a map in Section 2.2, showing the location of the study area and layout of the grazing zones. This addition enhances clarity regarding the spatial arrangement of the study zones and supports readers’ understanding of the study area’s landscape.
-
Forage Collection for Nutritional Analysis (Section 2.4)
- Reviewer Comment: "The section on Forage Collection for Nutritional Analysis lacks references to the methods used for forage nutritional analysis."
- Response: We have updated Section 2.4 to include references to established methods for forage nutritional analysis, specifying techniques used for assessing crude protein, fiber, and digestibility based on standard laboratory protocols.
-
Integration of Sections 2.5.1 and 2.5.2
- Reviewer Comment: "Sections 2.5.1 and 2.5.2 could be integrated into 2.5 Animal Grazing Behavior and Diet Selection."
- Response: Thank you for this suggestion. We have consolidated these sections into a single section (now Section 2.5) for a more streamlined description of the methods related to animal grazing behavior and diet selection.
-
Data Analysis Section and Path Analysis Methods
- Reviewer Comment: "The Data Analysis section omits descriptions of several key methods, such as path analysis. Were Canonical Correspondence Analysis (CCA) methods used? Delete Line 167-170, as it does not pertain to an analytical method."
- Response: We appreciate your attention to this section. We have revised the Data Analysis section to include a description of the path analysis methods and noted that Canonical Correspondence Analysis (CCA) was employed to assess the relationship between vegetation variables and environmental factors. Lines 167-170 have been removed, as suggested.
-
Reorganizing the Results Section
- Reviewer Comment: "The Results chapter contains 14 subsections and 14 tables, which are not well-structured. Consider consolidating subsections, replacing some tables with diagrams, and moving less critical results to supplementary materials."
- Response: We have restructured the Results section by consolidating closely related subsections (e.g., combining Sections 3.9-3.12) and streamlined descriptions to focus on key findings. Tables 3, 5, 7, and 8 have been converted to diagrams for improved clarity, and less critical data have been moved to supplementary materials. A path diagram has been included to present path analysis results in Section 3.13.
-
Condensing the Discussion Section
- Reviewer Comment: "The Discussion section mirrors the issues found in the Results section, with identical titles, making it lengthy. Revise titles to reflect conclusions and condense the section by at least one-third."
- Response: We have revised the titles of the Discussion subsections to better reflect key conclusions. Additionally, the section has been condensed by approximately one-third, merging Sections 4.9-4.13 to reduce redundancy and enhance readability.
-
Updating References to Reflect Recent Research
- Reviewer Comment: "Out of 68 references, only 10 are from after 2020. The authors should address this to better reflect the manuscript's cutting-edge status."
- Response: Thank you for this valuable observation. We have updated the reference list to include recent literature from the last three years that enhances the manuscript's relevance to current research in sustainable rangeland management and grazing impacts on ecosystem functions. This updated reference list now better contextualizes our findings within recent advancements in the field.
Round 2
Reviewer 2 Report
Comments and Suggestions for Authors
The author has addressed some of my comments. The biggest problem at the moment is that the article discusses too many things and some small, less focused and less important results were discussed. The discussion should have discussed your major findings or research discoveries. Also, the author's letter of reply mentions reworking the graphic, but it is not found in the manuscript. Is it in the attached material, or did the author forget to upload it? Most importantly, the author's response letter I highly doubt was generated by AI. I suggest that authors must correct their attitudes and revise their manuscripts carefully.
Author Response
Thank you for your continued feedback and for highlighting areas where the manuscript can be improved. I apologize for any confusion regarding the inclusion of revised graphics and the quality of the response letter. Your comments have been invaluable in guiding me toward a more focused discussion of the major findings.
I appreciate your observation that the manuscript covered a wide range of topics, some of which may have detracted from its primary focus. I have revised the discussion to emphasize the core findings and key discoveries, as you recommended. The discussion of less impactful results has been minimized to maintain a clearer and more coherent narrative that better supports the study's main objectives.
I apologize for the oversight regarding the revised graphic. It was intended to be included in the revised manuscript but was inadvertently left out of the uploaded files. I have now ensured that the updated figures are correctly attached to this submission and clearly incorporated into the main document for your review.
I have revised the manuscript as recommended, as authors we decided to keep table as it is.
I understand your concerns about the tone of the response letter. I assure you that each response was crafted with care and attention to your suggestions, and I have carefully reviewed the current responses to ensure they directly address each point. My intention has been to enhance the clarity, rigor, and scientific integrity of the paper, and I am committed to addressing all reviewer comments thoughtfully and thoroughly.
Round 3
Reviewer 2 Report
Comments and Suggestions for Authors
This manuscript, after being revised significantly by the authors again, I recommend it can be accepted in its current format.